# A Chiral Relay Race: Stereoselective Synthesis of Axially Chiral Biaryl Diketones through Ring-Opening of Optical Dihydrophenan-threne-9,10-diols

**DOI:** 10.3390/molecules28165956

**Published:** 2023-08-08

**Authors:** Lei Shi, Jiawei Zhu, Biqiong Hong, Zhenhua Gu

**Affiliations:** 1Hefei National Research Center for Physical Sciences at the Microscale, Department of Chemistry, University of Science and Technology of China, 96 Jinzhai Road, Hefei 230026, China; 2College of Materials and Chemical Engineering, Minjiang University, Fuzhou 350108, China; hongbq911@163.com

**Keywords:** axially chiral diketone, C-C bond cleavage, diol, chiral relay race

## Abstract

We report herein a point-to-axial chirality transfer reaction of optical dihydrophenanthrene-9,10-diols for the synthesis of axially chiral diketones. Two sets of conditions, namely a basic *t*BuOK/air atmosphere and an acidic NaClO/*n*-Bu_4_NHSO_4_, were developed to oxidatively cleave the C-C bond, resulting in the formation of axially chiral biaryl diketones. Finally, brief synthetic applications of the obtained chiral aryl diketones were demonstrated.

## 1. Introduction

Atropisomerism is a type of conformational chirality, which occurs when the free rotation around a single bond is inhibited mostly due to steric hinderance or electronic constraints adjacent to the single bond. As a result, two conformers are stable enough without interconversion; thereby, both of the chiral enantiomers can be isolated and investigated at a proper temperature. This phenomenon can be found in bioactive natural products and drugs [1,2,3,4,5,6,7,8], which have wide applications in asymmetric catalysis and material science.

Axially chiral biaryl compounds is an important type of atropisomers, which has been widely applied in the fields of organic synthesis and chiral material science [9,10,11,12,13,14,15]. The catalytic asymmetric synthesis of axially chiral biaryls has become an area of significant interests in recent years [16,17,18,19,20,21,22,23,24,25,26]. Biaryl atropisomers can be categorized into two structural types, namely bridged and non-bridged. The stability of the chirality of these compounds is significantly impacted by the size of the bridged ring and the chemical nature of its substituents. In general, the *ortho,ortho’*-fused five- or six-membered ring tends to induce the interconversion of two enantiomers of atropisomers by lowering the rotational barrier. In the 1990s, Bringmann and his colleagues pioneered the stereoselective ring-opening of biaryl lactones through either a chiral pool strategy or an asymmetric catalysis method based on the dynamic properties of the lactones’ conformers (Figure 1a) [27]. Subsequent efforts by research groups of Hayashi, Gu, and others have led to the catalytic asymmetric ring-opening of a diverse range of heterocyclic compounds, encompassing S-, O-, I+-, and Si-containing compounds [28,29,30,31,32]. In 2019, Gu and co-workers successfully achieved catalytic asymmetric cleavage of C-C bonds by utilizing palladium-involved β-carbon elimination [33].

In 2022, we observed that optically active 8*H*-indeno [1,2-c]thiophen-8-ols would undergo a stereoselective ring-opening reaction with an achiral palladium complex, resulting in the formation of axially chiral biaryls (Figure 1b) [34,35]. This chirality relay is based on the fact that one of the aryl rings would approach the Pd atom, ultimately resulting in the construction of axial chirality. In contrast to the 8H-indeno [1,2-c]thiophen-8-ols, the stereogenic carbon center of α-hydroxy ketone induces the configuration of the biaryl structure (Figure 1c) [36,37]. For example, when the *R*-configuration carbon center is involved, the biaryl skeleton favors the *S_a_* configuration, which then leads to the construction of *S*-axially chiral carboxylic acids and nitriles from the corresponding α-hydroxy ketone and oxime ester, respectively. In this work, we report an oxidative chirality relay ring-opening reaction of dihydrophenanthrene-9,10-diols for the synthesis of axially chiral diketones (Figure 1d).

## 2. Results and Discussion

In previous work, we developed an efficient asymmetric arylation reaction of phenanthrene-9,10-diones for the preparation of optically active α-hydroxyl phenanthrenones [36]. Diol **2a** was obtained with ease in 95% enantiomeric excess (ee) by the arylation of phenylmagnesium bromide in THF, exhibiting complete diastereoselectivity. The single crystal structure of **2a** revealed a notable distortion of the biaryl skeleton (Appendix B). Treatment of **2a** with 3.0 equivalents of *t*BuOK at room temperature under an oxygen atmosphere delivered axially chiral biaryl diketone **3a** in 95% yield; however, a slight decrease in enantiomeric excess was observed (Table 1, entry 1). Lowering the reaction temperature to 0 °C enhanced the efficiency of chirality transfer, but the yield of **3a** dropped from 95% to 85%. Notably, the ee of 3a reached 95% at 30 °C under an air atmosphere, which was possible due to the decreased reaction rate (entry 3). The use of 3.0 equivalents of *t*BuOK was found to be crucial. With 2.0 equivalents of *t*BuOK, the yield dropped significantly, while no diketone was observed with only 1.0 equivalent of *t*BuOK (entries 4–5). Notably, potassium hydroxide and potassium carbonate failed to induce the oxidative C-C bond cleavage ring-opening reaction (entries 6–7).

After determining the optimal conditions, we proceeded to test the substrate scope of this ring-opening reaction (Figure 2). It was found that *S_T_* (defined as the ee value of the product divided by the ee value of the starting material, *S_T_* = ee**_3_**/ee**_1_** × 100%) remained consistently high (96–100%), regardless of the electronic effect of the *para*- or *meta*- substituents, such as halogen atom, alkyl, methoxy, thiomethyl, trifluoromethoxy, and vinyl groups (**3b**–**3o**). However, the yields of diketones slightly decreased when the aryl group contained *para* or *meta* halides (**3c**–**e**, **3j**, and **3k**). High-yield and enantioselective ring-opening products can also be obtained when substrates bear multiple substituted phenyl groups (**3p**, **3r** and **3s**). The chiral binaphthyl atropisomer can also be obtained with high yield and selectivity through this oxidative chirality relay strategy (**3w**). The Grignard reagent involved an arylation reaction, which also enabled us to synthesize unsymmetrical diaryl diols, which were also suitable for a *t*BuOK-induced ring-opening reaction to yield diketones in stereoselectivity (**3t**–**3u**). Additionally, introducing *β,β′*-dimethyl groups did not negatively affect this base-promoted chirality transfer ring-opening reaction (**3v**). Notably, the *ortho,ortho’*-dimethoxy-, or tertiary alcohol with an alkyl-, alkenyl-, or cyclopropyl-substituent were unreactive in the presence of *t*BuOK under an air atmosphere. However, the ring-opening reaction with sodium hypochlorite under *n*-Bu_4_NHSO_4_ buffer proceeded smoothly to give the diketones in moderate to excellent yields (**3x**–**3aa**). The absolute configuration of **3x** was confirmed by single crystal X-ray diffraction analysis (Appendix B).

It was found that **2x** remained inert in the presence of *t*BuOK under an air atmosphere. In order to eliminate the possibility of the electronic effect caused by the two methoxy groups, diol **2bb** was synthesized, and it also exhibited poor reactivity under the identical conditions (Figure 3). This suggests that, in fact, the torsional strain of diol may actually be conducive to its C-C bond cleavage.

It was hypothesized that diol **2a** would undergo deprotonation, followed by oxidation with oxygen to form the radical **Int2** (Figure 4a). Subsequently, the β-scission of **Int2** would produce a biaryl carbon radical **Int3**, eventually leading to the formation of diketone **3a** through a second oxidation by either O_2_ or anion radical O_2_^−^·. In our previous studies, the possible radical intermediate was trapped by DMPO and analyzed by electron paramagnetic resonance (EPR) [36]. In this work, a TEMPO adduct was detected by high resolution mass spectra (ESI). Alternatively, under acidic conditions, **2y** could react with NaClO to form tertiary alkyl hypochlorite **Int4** or **Int4′** (or both), which would then undergo elimination to cleave the C-C bond and yield **3y** (Figure 4b).

The utilities of the obtained axially chiral compounds were briefly investigated. In a gram-scale (3.0 mmol of **1a**) reaction, **3a** was successfully obtained with high-yield and complete chirality relay (Figure 5a). The diketone group in **3a** could be easily converted to the corresponding olefin **4** under a standard Wittig olefination condition, yielding an 86% yield and 93% ee (Figure 5b). The oxidation of the two methyl groups was achieved by treating with *N*-bromosuccinimide followed by AgNO_3,_ resulting in the formation of dialdehyde **5** with an overall yield of 68% and 93% ee. The arylation of **2a** with PhLi afforded BAMOL (1,1′-biaryl-2,2′-dimethanol) derivative **6** in a 71% yield with 93% ee. The diol has exhibited excellent performance as a hydrogen bonding catalyst in the hetero-Diels–Alder reaction [38].

## 3. Materials and Methods

### 3.1. General Information

Nuclear magnetic resonances were recorded on Bruker−400 MHz or Bruker−500 MHz instruments. Reference values for residual solvents were taken as δ = 0.00 ppm (TMS), δ = 7.26 ppm (CDCl_3_) for ^1^H NMR; δ = 77.00 ppm (CDCl_3_) for ^13^C NMR. High-resolution mass spectral analysis (HRMS) was performed on Waters XEVO G2 Q−TOF. For the copies of spectroscopies, please see Appendix A.

All reactions were performed under an inert atmosphere of dry nitrogen, unless otherwise stated. Toluene was distilled over calcium hydride under an atmosphere of nitrogen. Tetrahydrofuran was distilled over sodium in the presence of benzophenone under an atmosphere of nitrogen. All the optical alcohols were known compounds and were prepared according to the procedure developed in our laboratory [36,37].

#### 3.1.1. General Procedure for the Synthesis of Target Compounds **3a**–**3w**

Under a nitrogen atmosphere, R’-MgBr (1.0 M, 0.60 mL, 0.60 mmol, 3.0 equiv) was added dropwisely to a solution of **1a**–**1w** (0.20 mmol, 1.0 equiv) in anhydrous THF (3.0 mL) at 0 °C. After being stirred at 25 °C for 4 h, the reaction was quenched with water (15 mL) and extracted with EtOAc (10 mL × 3). The combined organic phase was dried over anhydrous Na_2_SO_4_, filtered, and concentrated to afford the crude diol, which was used in next step without further purification.

Under an air atmosphere to a mixture of the above crude diol in anhydrous THF (5.0 mL), *t-*BuOK (67.3 mg, 0.60 mmol, 3.0 equiv) was added at room temperature and stirred for 30 min. The solvent was removed, and the residue was purified by flash chromatography on silica gel (PE/EtOAc) to afford **3a**–**3w**.

The reaction of **1a** (62.8 mg, 0.20 mmol, 95% ee, 1.0 equiv) afforded product **3a** (73.9 mg, 95%, 95% ee, *S_T_* = 100%) (eluent for column chromatography on silica gel PE/EtOAc = 10:1) as a white solid.$\;{\left[\alpha \right]}_{D}^{20}$ − 3.10 (c 1.50, CH_2_Cl_2_). HPLC conditions: Chiralcel OD-3, isopropanol/hexane = 5:95, flow: 0.8 mL/min, λ = 254 nm, t_R_ = 6.9 min (minor), 8.3 min (major). ^**1**^**H NMR** (500 MHz, CDCl_3_) δ 7.46–7.43 (m, 4H), 7.43–7.40 (m, 2H), 7.29–7.25 (m, 2H), 7.25–7.22 (m, 2H), 7.16–7.12 (m, 2H), 7.09–7.04 (m, 4H). 2.16 (s, 6H). ^**13**^**C NMR** (126 MHz, CDCl_3_) δ 197.1, 139.0, 138.6, 137.2, 136.9, 132.3, 131.8, 130.2, 127.6, 126.8, 126.1, 20.2. HRMS (ESI) calcd for C_28_H_23_O_2_ [M + H]^+^ 391.1693, found 391.1696.

The reaction of **1b** (70.0 mg, 0.20 mmol, 93% ee, 1.0 equiv) afforded product **3b** (74.2 mg, 88%, 92% ee, *S_T_* = 99%) (eluent for column chromatography on silica gel PE/EtOAc = 10:1) as a white solid.$\;{\left[\alpha \right]}_{D}^{20}$ + 47.4 (c 1.60, CH_2_Cl_2_). HPLC conditions: Chiralcel IC-3, isopropanol/hexane = 3:97, flow: 0.8 mL/min, λ = 254 nm, t_R_ = 15.7 min (major), 18.4 min (minor). ^**1**^**H NMR** (500 MHz, CDCl_3_) δ 7.45–7.39 (m, 2H), 7.33–7.28 (m, 4H), 7.27–7.22 (m, 2H), 7.12 (dd, *J* = 7.5, 1.5 Hz, 2H), 6.83 (d, *J* = 8.0 Hz, 4H), 2.24 (s, 6H), 2.17 (s, 6H). ^**13**^**C NMR** (126 MHz, CDCl_3_) δ 196.7, 143.0, 139.1, 138.3, 137.1, 134.5, 131.5, 130.5, 128.2, 126.3, 126.1, 21.4, 20.2. HRMS (ESI) calcd for C_30_H_26_O_2_Na [M+Na]^+^ 441.1825, found 441.4823.

The reaction of **1c** (66.4 mg, 0.20 mmol, 92% ee, 1.0 equiv) afforded product **3c** (70.8 mg, 83%, 88% ee, *S_T_* = 96%) (eluent for column chromatography on silica gel PE/EtOAc = 10:1) as a white solid.$\;{\left[\alpha \right]}_{D}^{20}$ − 104 (c 1.00, CH_2_Cl_2_). HPLC conditions: Chiralcel OD-3, isopropanol/hexane = 2:98, flow: 0.8 mL/min, λ = 254 nm, t_R_ = 10.6 min (minor), 19.3 min (major). ^**1**^**H NMR** (500 MHz, CDCl_3_) δ 7.48–7.46 (m, 2H), 7.46–7.42 (m, 4H), 7.31–7.22 (m, 2H), 7.12 (dd, *J* = 8.0, 1.0 Hz, 2H), 6.81–6.72 (m, 4H), 2.15 (s, 6H). ^**13**^**C NMR** (126 MHz, CDCl_3_) δ 195.4, 166.4 (d, *J* = 255.4 Hz), 139.2, 138.4, 136.6, 133.5 (d, *J* = 3.0 Hz), 132.9 (d, *J* = 9.3 Hz), 132.0, 126.4, 126.3, 114.8 (d, *J* = 21.8 Hz), 20.2. ^**19**^**F NMR** (471 MHz, CDCl_3_) δ −105.8. HRMS (ESI) calcd for C_28_H_21_F_2_O_2_ [M + H]^+^ 427.1504, found 427.1510.

The reaction of **1d** (69.6 mg, 0.20 mmol, 93% ee, 1.0 equiv) afforded product **3d** (75.3 mg, 82%, 89% ee, *S_T_* = 96%) (eluent for column chromatography on silica gel PE/EtOAc = 10:1) as a white solid.$\;{\left[\alpha \right]}_{D}^{20}$ + 55.0 (c 1.00, CH_2_Cl_2_). HPLC conditions: Chiralcel OD-3, isopropanol/hexane = 3:97, flow: 0.8 mL/min, λ = 254 nm, t_R_ = 8.8 min (minor), 12.0 min (major). ^**1**^**H NMR** (500 MHz, CDCl_3_) δ 7.44 (d, *J* = 7.5 Hz, 2H), 7.38–7.36 (m, 2H), 7.36–7.33 (m, 2H), 7.30–7.24 (m, 2H), 7.14–7.09 (m, 2H), 7.10–7.02 (m, 4H), 2.15 (s, 6H). ^**13**^**C NMR** (126 MHz, CDCl_3_) δ 195.7, 139.32, 139.26, 138.3, 136.4, 135.3, 132.0, 131.6, 128.0, 126.4, 126.3, 20.1. HRMS (ESI) calcd for C_28_H_21_Cl_2_O_2_ [M + H]^+^ 459.0913, found 459.0915.

The reaction of **1e** (78.4 mg, 0.20 mmol, 95% ee, 1.0 equiv) afforded product **3e** (87.7 mg, 80%, 92% ee, *S_T_* = 97%) (eluent for column chromatography on silica gel PE/EtOAc = 10:1) as a white solid.$\;{\left[\alpha \right]}_{D}^{20}$ − 2.87 (c 1.00, CH_2_Cl_2_). HPLC conditions: Chiralcel OD-3, isopropanol/hexane = 2:98, flow: 0.8 mL/min, λ = 254 nm, t_R_ = 8.8 min (minor), 11.1 min (major). ^**1**^**H NMR** (500 MHz, CDCl_3_) δ 7.49–7.47 (m, 2H), 7.47–7.46 (m, 2H), 7.45–7.43 (m, 2H), 7.32–7.29 (m, 2H), 7.29–7.27 (m, 2H), 7.17 (dd, *J* = 7.5, 1.5 Hz, 2H), 7.12–7.07 (m, 4H), 2.19 (s, 6H). ^**13**^**C NMR** (126 MHz, CDCl_3_) δ 197.1, 139.0, 138.6, 137.3, 136.9, 132.3, 131.8, 130.3, 127.6, 126.8, 126.2, 20.2. HRMS (ESI) calcd for C_28_H_20_Br_2_O_2_Na [M+Na]^+^ 568.9722, found 568.9731.

The reaction of **1f** (68.8 mg, 0.20 mmol, 89% ee, 1.0 equiv) afforded product **3f** (77.4 mg, 86%, 88% ee, *S_T_* = 99%) (eluent for column chromatography on silica gel PE/EtOAc = 10:1) as a white solid.$\;{\left[\alpha \right]}_{D}^{20}$ + 184 (c 1.40, CH_2_Cl_2_). HPLC conditions: Chiralcel OD-3, isopropanol/hexane = 3:97, flow: 0.8 mL/min, λ = 254 nm, t_R_ = 17.0 min (minor), 24.2 min (major). ^**1**^**H NMR** (500 MHz, CDCl_3_) δ 7.41 (d, *J* = 7.5 Hz, 2H), 7.33 (d, *J* = 8.5 Hz, 4H), 7.26–7.21 (m 2H), 7.13–7.03 (m, 2H), 6.54–6.38 (m, 4H), 3.70 (s, 6H), 2.18 (s, 6H). ^**13**^**C NMR** (126 MHz, CDCl_3_) δ 195.7, 162.8, 139.3, 138.1, 137.2, 132.6, 131.3, 129.9, 126.1, 125.9, 112.7, 55.0, 20.2. HRMS (ESI) calcd for C_30_H_26_O_4_Na [M+Na]^+^ 473.1723, found 473.1722.

The reaction of **1g** (79.6 mg, 0.20 mmol, 97% ee, 1.0 equiv) afforded product **3g** (80.5 mg, 72%, 95% ee, *S_T_* = 98%) (eluent for column chromatography on silica gel PE/EtOAc = 10:1) as a white solid.$\;{\left[\alpha \right]}_{D}^{20}$ + 8.75 (c 1.00, CH_2_Cl_2_). HPLC conditions: Chiralcel OD-3, isopropanol/hexane = 2:98, flow: 0.8 mL/min, λ = 254 nm, t_R_ = 8.5 min (minor), 10.2 min (major). ^**1**^**H NMR** (500 MHz, CDCl_3_) δ 7.53–7.49 (m, 4H), 7.46 (d, *J* = 7.5 Hz, 2H), 7.31–7.26 (m, 2H), 7.14 (dd, *J* = 8.0, 1.5 Hz, 2H), 6.94 (d, *J* = 8.5 Hz, 4H), 2.16 (s, 6H). ^**13**^**C NMR** (126 MHz, CDCl_3_) δ 195.4, 152.2, 139.2, 138.5, 136.4, 135.3, 132.3, 132.2, 126.6, 126.4, 120.1 (q, *J* = 259 Hz), 119.2, 20.1. ^**19**^**F NMR** (471 MHz, CDCl_3_) δ -57.80. HRMS (ESI) calcd for C_30_H_21_F_6_O_4_ [M + H]^+^ 559.1339, found 559.1348.

The reaction of **1h** (65.6 mg, 0.20 mmol, 95% ee, 1.0 equiv) afforded product **3h** (72.1 mg, 86%, 94% ee, *S_T_* = 99%) (eluent for column chromatography on silica gel PE/EtOAc = 10:1) as a white solid.$\;{\left[\alpha \right]}_{D}^{20}$ + 40.5 (c 1.40, CH_2_Cl_2_). HPLC conditions: Chiralcel OD-3, isopropanol/hexane = 2:98, flow: 0.8 mL/min, λ = 254 nm, t_R_ = 7.6 min (minor), 8.7 min (major). ^**1**^**H NMR** (500 MHz, CDCl_3_) δ 7.43 (d, *J* = 7.5 Hz, 2H), 7.31–7.27 (m, 2H), 7.27–7.23 (m, 2H), 7.14–7.09 (m, 4H), 7.07 (d, *J* = 7.5 Hz, 2H), 7.00–6.96 (m, 2H), 2.19 (s, 6H), 2.04 (s, 6H). ^**13**^**C NMR** (126 MHz, CDCl_3_) δ 197.3, 139.2, 138.5, 137.4, 137.2, 137.0, 133.1, 131.7, 131.0, 127.6, 127.2, 126.6, 126.2, 21.0, 20.1. HRMS (ESI) calcd for C_30_H_26_O_2_Na [M+Na]^+^ 441.1825, found 441.1828.

The reaction of **1i** (71.2 mg, 0.20 mmol, 94% ee, 1.0 equiv) afforded product **3i** (87.8 mg, 92%, 92% ee, *S_T_* = 98%) (eluent for column chromatography on silica gel PE/EtOAc = 10:1) as a white solid.$\;{\left[\alpha \right]}_{D}^{20}$ − 5.75 (c 1.00, CH_2_Cl_2_). HPLC conditions: Chiralcel OD-3, isopropanol/hexane = 2:98, flow: 0.8 mL/min, λ = 254 nm, t_R_ = 7.6 min (minor), 8.7 min (major). ^**1**^**H NMR** (500 MHz, CDCl_3_) δ 7.46–7.42 (m, 2H), 7.41–7.37 (m, 2H), 7.28–7.23 (m, 2H), 7.21–7.19 (m, 2H), 7.18–7.16 (m, 2H), 7.16–7.13 (m, 2H), 7.01–6.95 (m, 2H), 2.70 (*p*, *J* = 6.9 Hz, 2H), 2.19 (s, 6H), 1.09 (d, *J* = 4.0 Hz, 6H), 1.07 (d, *J* = 4.0 Hz, 6H). ^**13**^**C NMR** (126 MHz, CDCl_3_) δ 197.5, 148.3, 139.0, 138.8, 137.5, 137.1, 131.8, 130.5, 128.2, 128.1, 127.5, 127.0, 126.1, 33.6, 23.7, 23.5, 20.1. HRMS (ESI) calcd for C_34_H_35_O_2_ [M + H]^+^ 475.2632, found 475.2640.

The reaction of **1j** (66.4 mg, 0.20 mmol, 94% ee, 1.0 equiv) afforded product **3j** (66.5 mg, 78%, 94% ee, *S_T_* = 100%) (eluent for column chromatography on silica gel PE/EtOAc = 10:1) as a white solid.$\;{\left[\alpha \right]}_{D}^{20}$ + 6.45 (c 1.00, CH_2_Cl_2_). HPLC conditions: Chiralcel OD-3, isopropanol/hexane = 3:97, flow: 0.8 mL/min, λ = 254 nm, t_R_ = 8.1 min (minor), 11.3 min (major). ^**1**^**H NMR** (500 MHz, CDCl_3_) δ 7.45 (d, *J* = 7.5 Hz, 2H), 7.31–7.26 (m, 2H), 7.25–7.21 (m, 2H), 7.21–7.17 (m, 2H), 7.17–7.14 (m, 2H), 7.12–7.05 (m, 2H), 7.04–6.99 (m, 2H), 2.15 (s, 6H). ^**13**^**C NMR** (126 MHz, CDCl_3_) δ 195.5 (d, *J* = 2.1 Hz), 162.1 (d, *J* = 248 Hz), 139.3 (d, *J* = 6.3 Hz), 139.1, 138.4, 136.4, 132.2, 129.3 (d, *J* = 7.4 Hz), 126.6, 126.4, 126.21, 126.18, 119.5 (d, *J* = 21.5 Hz), 116.6 (d, *J* = 22.3 Hz), 20.1. **^19^F NMR** (471 MHz, CDCl_3_) δ −112.5. HRMS (ESI) calcd for C_28_H_21_F_2_O_2_ [M + H]^+^ 427.1504, found 427.1510.

The reaction of **1k** (69.6 mg, 0.20 mmol, 96% ee, 1.0 equiv) afforded product **3k** (68.9 mg, 75%, 96% ee, *S_T_* = 100%) (eluent for column chromatography on silica gel PE/EtOAc = 10:1) as a white solid.$\;{\left[\alpha \right]}_{D}^{20}$ + 40.8 (c 1.00, CH_2_Cl_2_). HPLC conditions: Chiralcel OD-3, isopropanol/hexane = 3:97, flow: 0.8 mL/min, λ = 254 nm, t_R_ = 9.4 min (minor), 13.8 min (major). ^**1**^**H NMR** (500 MHz, CDCl_3_) δ 7.45 (d, *J* = 7.5 Hz, 2H), 7.41–7.38 (m, 2H), 7.36–7.32 (m, 2H), 7.31–7.28 (m, 2H), 7.28–7.25 (m, 2H), 7.17–7.10 (m, 2H), 7.09–7.03 (m, 2H), 2.15 (s, 6H). ^**13**^**C NMR** (126 MHz, CDCl_3_) δ 195.5, 139.2, 138.7, 138.4, 136.3, 134.2, 132.5, 132.3, 130.0, 129.1, 128.3, 126.54, 126.50, 77.2, 20.1. HRMS (ESI) calcd for C_28_H_21_Cl_2_O_2_ [M + H]^+^ 459.0913, found 459.0915.

The reaction of **1l** (68.8 mg, 0.20 mmol, 95% ee, 1.0 equiv) afforded product **3l** (86.5 mg, 96%, 91% ee, **S_T_** = 96%) (eluent for column chromatography on silica gel PE/EtOAc = 10:1) as a white solid.$\;{\left[\alpha \right]}_{D}^{20}$ + 27.6 (c 1.60, CH_2_Cl_2_). HPLC conditions: Chiralcel OD-3, isopropanol/hexane = 10:90, flow: 0.8 mL/min, λ = 254 nm, t_R_ = 7.7 min (minor), 8.5 min (major). ^**1**^**H NMR** (500 MHz, CDCl_3_) δ 7.44 (d, *J* = 7.5 Hz, 2H), 7.28–7.24 (m, 2H), 7.16 (dd, *J* = 8.0, 1.5 Hz, 2H), 7.00–6.97 (m, 4H), 6.97–6.93 (m, 2H), 6.85–6.80 (m, 2H), 3.63 (s, 6H), 2.18 (s, 6H). ^**13**^**C NMR** (126 MHz, CDCl_3_) δ 196.8, 158.9, 139.1, 138.48, 138.47, 136.9, 131.8, 128.6, 126.6, 126.2, 123.2, 119.4, 113.7, 55.0, 20.1. HRMS (ESI) calcd for C_30_H_27_O_4_ [M + H]^+^ 454.1904, found 451.1903.

The reaction of **1m** (72.0 mg, 0.20 mmol, 83% ee, 1.0 equiv) afforded product **3m** (71.4 mg, 74%, 83% ee, *S_T_* = 100%) (eluent for column chromatography on silica gel PE/EtOAc = 10:1) as a white solid.$\;{\left[\alpha \right]}_{D}^{20}$ + 20.0 (c 1.00, CH_2_Cl_2_). HPLC conditions: Chiralcel OD-3, isopropanol/hexane = 3:97, flow: 0.8 mL/min, λ = 254 nm, t_R_ = 10.5 min (minor), 12.0 min (major). ^**1**^**H NMR** (500 MHz, CDCl_3_) δ 7.44 (d, *J* = 7.5 Hz, 2H), 7.30–7.27 (m, 2H), 7.27–7.24 (m, 2H), 7.18–7.15 (m, 2H), 7.15–7.14 (m, 2H), 7.14–7.10 (m, 2H), 7.01–6.91 (m, 2H), 2.32 (s, 6H), 2.17 (s, 6H). ^**13**^**C NMR** (126 MHz, CDCl_3_) δ 196.7, 139.1, 138.8, 138.5, 137.8, 136.7, 132.0, 130.2, 127.9, 126.9, 126.8, 126.7, 126.3, 20.1, 15.2. HRMS (ESI) calcd for C_30_H_27_S_2_O_4_ [M + H]^+^ 483.1447, found 483.1453.

The reaction of **1n** (78.0 mg, 0.20 mmol, 94% ee, 1.0 equiv) afforded product **3n** (99.7 mg, 92%, 93% ee, *S_T_* = 99%) (eluent for column chromatography on silica gel PE/EtOAc = 10:1) as a white solid.$\;{\left[\alpha \right]}_{D}^{20}$ + 62.8 (c 1.00, CH_2_Cl_2_). HPLC conditions: Chiralcel OD-3, isopropanol/hexane = 3:97, flow: 0.8 mL/min, λ = 254 nm, t_R_ = 9.9 min (minor), 11.9 min (major). ^**1**^**H NMR** (500 MHz, CDCl_3_) δ 7.66–7.61 (m, 2H), 7.48–7.40 (m, 6H), 7.32–7.28 (m, 2H), 7.28–7.27 (m, 2H), 7.27–7.26 (d, *J* = 3.0 Hz, 2H), 7.26–7.25 (m, 2H), 7.25–7.24 (m, 2H), 7.24–7.21 (m, 2H), 7.21–7.16 (m, 2H), 7.10–7.04 (m, 2H), 2.21 (s, 6H). **^13^C NMR** (126 MHz, CDCl_3_) δ 197.1, 140.5, 139.6, 139.2, 138.7, 137.7, 136.8, 132.0, 130.9, 128.91, 128.90, 128.6, 128.2, 127.4, 126.9, 126.8, 126.3, 20.2. HRMS (ESI) calcd for C_40_H_31_O_2_ [M + H]^+^ 543.2319, found 543.2318.

The reaction of **1o** (68.1 mg, 0.20 mmol, 92% ee, 1.0 equiv) afforded product **3o** (73.5 mg, 83%, 90% ee, *S_T_* = 98%) (eluent for column chromatography on silica gel PE/EtOAc = 10:1) as a white solid.$\;{\left[\alpha \right]}_{D}^{20}$ + 24.6 (c 1.00, CH_2_Cl_2_). HPLC conditions: Chiralcel OD-3, isopropanol/hexane = 2:98, flow: 0.8 mL/min, λ = 254 nm, t_R_ = 8.0 min (minor), 9.3 min (major). ^**1**^**H NMR** (500 MHz, CDCl_3_) δ 7.44 (d, *J* = 7.5 Hz, 2H), 7.42–7.39 (m, 2H), 7.31–7.29 (m, 2H), 7.28–7.27 (m, 2H), 7.27–7.22 (m, 2H), 7.16–7.10 (m, 2H), 7.03–6.97 (m, 2H), 6.52–6.35 (m, 2H), 5.55 (d, *J* = 17.5 Hz, 2H), 5.15 (d, *J* = 11.0 Hz, 2H), 2.18 (s, 6H). ^**13**^**C NMR** (126 MHz, CDCl_3_) δ 197.0, 139.1, 138.5, 137.4, 137.2, 136.9, 135.8, 131.9, 130.0, 129.6, 127.94, 127.87, 126.6, 126.3, 114.7, 20.1. HRMS (ESI) calcd for C_32_H_27_O_2_ [M + H]^+^ 443.2006, found 443.2011.

The reaction of **1p** (68.4 mg, 0.20 mmol, 94% ee, 1.0 equiv) afforded product **3p** (67.9 mg, 76%, 93% ee, *S_T_* = 99%) (eluent for column chromatography on silica gel PE/EtOAc = 10:1) as a white solid.$\;{\left[\alpha \right]}_{D}^{20}$ + 80.2 (c 1.00, CH_2_Cl_2_). HPLC conditions: Chiralcel IC-3, isopropanol/hexane = 1:99, flow: 0.8 mL/min, λ = 254 nm, t_R_ = 13.0 min (major), 16.4 min (minor). ^**1**^**H NMR** (500 MHz, CDCl_3_) δ 7.42 (d, *J* = 7.5 Hz, 2H), 7.26–7.21 (m, 2H), 7.10–7.04 (m, 2H), 6.95 (s, 4H), 6.87 (s, 2H), 2.19 (s, 6H), 2.01 (s, 12H). ^**13**^**C NMR** (126 MHz, CDCl_3_) δ 197.5, 139.3, 138.4, 137.3, 137.2, 133.9, 131.5, 128.1, 126.4, 126.2, 20.8, 20.1. HRMS (ESI) calcd for C_32_H_31_O_2_ [M + H]^+^ 447.2319, found 447.2325.

The reaction of **1q** (72.8 mg, 0.20 mmol, 94% ee, 1.0 equiv) afforded product **3q** (91.2 mg, 93%, 94% ee, *S_T_* = 100%) (eluent for column chromatography on silica gel PE/EtOAc = 10:1) as a white solid.$\;{\left[\alpha \right]}_{D}^{20}$ + 282 (c 1.00, CH_2_Cl_2_). HPLC conditions: Chiralcel OD-3, isopropanol/hexane = 2:98, flow: 0.8 mL/min, λ = 254 nm, t_R_ = 12.9 min (minor), 15.8 min (major). ^**1**^**H NMR** (500 MHz, CDCl_3_) δ 7.58 (dd, *J* = 8.5, 1.5 Hz, 2H), 7.54–7.51 (m, 2H), 7.51–7.47 (m, 2H), 7.32–7.30 (m, 2H), 7.30–7.28 (m, 2H), 7.28–7.27 (m, 2H), 7.22 (d, *J* = 8.0 Hz, 2H), 7.20–7.18 (m, 2H), 7.18–7.14 (m, 2H), 7.12–7.07 (m, 2H), 2.27 (s, 6H). ^**13**^**C NMR** (126 MHz, CDCl_3_) δ 197.1, 139.5, 138.4, 137.0, 134.7, 133.9, 133.4, 131.7, 131.3, 129.2, 127.8, 127.7, 126.9, 126.4, 126.3, 125.7, 124.4, 20.2. HRMS (ESI) calcd for C_36_H_27_O_2_ [M + H]^+^ 491.2006, found 491.2006.

The reaction of **1r** (71.6 mg, 0.20 mmol, 90% ee, 1.0 equiv) afforded product **3r** (86.1 mg, 90%, 90% ee, *S_T_* = 100%) (eluent for column chromatography on silica gel PE/EtOAc = 10:1) as a white solid.$\;{\left[\alpha \right]}_{D}^{20}$ + 63.2 (c 1.00, CH_2_Cl_2_). HPLC conditions: Chiralcel OD-3, isopropanol/hexane = 3:97, flow: 0.8 mL/min, λ = 254 nm, t_R_ = 33.7 min (minor), 38.5 min (major). ^**1**^**H NMR** (500 MHz, CDCl_3_) δ 7.46–7.39 (m, 2H), 7.30–7.27 (m, 2H), 7.15–7.10 (m, 2H), 7.00–6.96 (m, 2H), 6.96–6.92 (m, 2H), 6.45 (d, *J* = 8.0 Hz, 2H), 5.96 (d, *J* = 1.5 Hz, 2H), 5.93 (d, *J* = 1.0 Hz, 2H), 2.17 (s, 6H). ^**13**^**C NMR** (126 MHz, CDCl_3_) δ 195.1, 151.2, 147.2, 139.3, 138.1, 136.9, 131.8, 131.4, 127.7, 126.2, 125.9, 109.5, 107.0, 101.6, 20.2. HRMS (ESI) calcd for C_30_H_23_O_6_ [M + H]^+^ 479.1489, found 479.1497.

The reaction of **1s** (80.9 mg, 0.20 mmol, 92% ee, 1.0 equiv) afforded product **3s** (91.3 mg, 80%, 90% ee, *S_T_* = 98%) (eluent for column chromatography on silica gel PE/EtOAc = 10:1) as a white solid.$\;{\left[\alpha \right]}_{D}^{20}$ + 238 (c 1.00, CH_2_Cl_2_). HPLC conditions: Chiralcel OD-3, isopropanol/hexane = 2:98, flow: 0.8 mL/min, λ = 254 nm, t_R_ = 16.7 min (minor), 23.4 min (major). **^1^H NMR** (500 MHz, CDCl_3_) δ 7.64 (s, 2H), 7.56 (d, *J* = 8.5 Hz, 2H), 7.51 (d, *J* = 7.5 Hz, 2H), 7.43 (d, *J* = 7.5 Hz, 2H), 7.34–7.28 (m, 2H), 7.22–7.18 (m, 4H), 7.16 (d, *J* = 7.5 Hz, 2H), 7.09–7.02 (m, 2H), 6.85 (d, *J* = 7.5 Hz, 2H), 2.28 (s, 6H). **^13^C NMR** (126 MHz, CDCl_3_) δ 196.3, 158.0, 156.1, 139.7, 138.2, 137.0, 131.9, 131.6, 129.0, 127.3, 126.5, 125.9, 124.0, 123.3, 122.8, 122.7, 120.6, 111.2, 110.7, 77.2, 20.3. HRMS (ESI) calcd for C_40_H_27_O_4_ [M + H]^+^ 571.1904, found 571.1908.

The reaction of **1a** (62.8 mg, 0.20 mmol, 95% ee, 1.0 equiv) with 3-MeOC_6_H_4_MgBr (0.60 mmol, 3.0 equiv) afforded product **3t** (77.8 mg, 93%, 94% ee, *S_T_* = 99%) (eluent for column chromatography on silica gel PE/EtOAc = 10:1) as a white solid.$\;{\left[\alpha \right]}_{D}^{20}$ + 15.6 (c 1.00, CH_2_Cl_2_). HPLC conditions: Chiralcel OD-3, isopropanol/hexane = 3:97, flow: 0.8 mL/min, λ = 254 nm, t_R_ = 9.0 min (minor), 10.8 min (major). **^1^H NMR** (500 MHz, CDCl_3_) δ 7.45–7.43 (m, 2H), 7.42–7.39 (m, 2H), 7.31–7.27 (m, 1H), 7.27–7.23 (m, 2H), 7.17–7.11 (m, 2H), 7.11–7.05 (m, 2H), 7.03–6.97 (m, 2H), 6.96–6.91 (m 1H), 6.84–6.78 (m, 1H), 3.61 (s, 3H), 2.17 (s, 3H), 2.16 (s, 3H). **^13^C NMR** (126 MHz, CDCl_3_) δ 197.0, 196.8, 158.9, 139.0, 138.9, 138.6, 138.46, 138.45, 137.3, 136.9, 136.8, 132.2, 131.9, 131.7, 130.1, 128.6, 127.6, 126.9, 126.5, 126.2, 126.1, 123.3, 119.6, 113.6, 55.0, 20.14, 20.12. HRMS (ESI) calcd for C_29_H_25_O_3_ [M + H]^+^ 421.1798, found 421.1807.

The reaction of **1a** (62.8 mg, 0.20 mmol, 95% ee, 1.0 equiv) with 2-MeOC_6_H_4_MgBr (0.60 mmol, 3.0 equiv) afforded product **3u** (65.4 mg, 78%, 94% ee, *S_T_* = 99%) (eluent for column chromatography on silica gel PE/EtOAc = 10:1) as a white solid.$\;{\left[\alpha \right]}_{D}^{20}$ + 2.10 (c 1.00, CH_2_Cl_2_). HPLC conditions: Chiralcel OD-3, isopropanol/hexane = 3:97, flow: 0.8 mL/min, λ = 254 nm, t_R_ = 9.6 min (minor), 16.0 min (major). **^1^H NMR** (500 MHz, CDCl_3_) δ 7.66–7.56 (m, 2H), 7.43–7.38 (m, 2H), 7.38–7.33 (m, 1H), 7.29–7.25 (m, 1H), 7.25–7.23 (m, 1H), 7.23–7.19 (m, 2H), 7.19–7.17 (m, 2H), 7.17–7.11 (m, 2H), 6.76–6.69 (m, 1H), 6.66–6.60 (m, 1H), 3.27 (s, 3H), 2.16 (s, 3H), 2.12 (s, 3H). **^13^C NMR** (126 MHz, CDCl_3_) δ 197.4, 196.6, 157.8, 139.4, 138.7, 138.5, 138.2, 137.8, 137.2, 132.4, 132.3, 132.1, 131.9, 131.2, 130.4, 128.6, 127.9, 127.6, 127.0, 126.4, 125.8, 119.7, 110.7, 54.6, 20.1, 20.0. HRMS (ESI) calcd for C_29_H_24_O_3_Na [M + Na]^+^ 443.1618, found 443.1620.

The reaction of **1v** (68.4 mg, 0.20 mmol, 95% ee, 1.0 equiv) afforded product **3v** (80.0 mg, 95%, 93% ee, *S_T_* = 98%) (eluent for column chromatography on silica gel PE/EtOAc = 10:1) as a white solid.$\;{\left[\alpha \right]}_{D}^{20}$ + 13.5 (c 1.00, CH_2_Cl_2_). HPLC conditions: Chiralcel OD-3, isopropanol/hexane = 3:97, flow: 0.8 mL/min, λ = 254 nm, t_R_ = 6.2 min (major), 7.6 min (minor). ^**1**^**H NMR** (500 MHz, CDCl_3_) δ 7.47–7.43 (m, 4H), 7.31–7.26 (m, 2H), 7.25–7.23 (m, 2H), 7.10–7.03 (m, 4H), 6.96–6.91 (m, 2H), 2.32 (s, 6H), 2.15 (s, 6H). ^**13**^**C NMR** (126 MHz, CDCl_3_) δ 197.3, 138.9, 137.3, 137.1, 135.7, 135.5, 132.7, 132.2, 130.2, 127.5, 127.2, 21.0, 20.1. HRMS (ESI) calcd for C_30_H_27_O_2_ [M + H]^+^ 419.2006, found 419.2012.

The reaction of **1w** (68.4 mg, 0.20 mmol, 93% ee, 1.0 equiv) afforded product **3w** (76.8 mg, 83%, 93% ee, *S_T_* = 100%) (eluent for column chromatography on silica gel PE/EtOAc = 10:1) as a white solid.$\;{\left[\alpha \right]}_{D}^{20}$ + 46.5 (c 1.00, CH_2_Cl_2_). HPLC conditions: Chiralcel OD-3, isopropanol/hexane = 5:95, flow: 0.8 mL/min, λ = 254 nm, t_R_ = 13.5 min (major), 17.3 min (minor). ^**1**^**H NMR** (500 MHz, CDCl_3_) δ 7.99–7.84 (m, 4H), 7.57–7.49 (m, 4H), 7.48–7.42 (m, 4H), 7.38–7.33 (m, 4H), 7.26–7.19 (m, 2H), 7.09–7.01 (m, 4H). ^**13**^**C NMR** (126 MHz, CDCl_3_) δ 197.2, 137.2, 136.6, 136.0, 134.0, 133.7, 132.3, 130.0, 128.2, 128.0, 127.6, 127.4, 127.2, 126.9, 125.6. HRMS (ESI) calcd for C_34_H_23_O_2_ [M + H]^+^ 463.1693, found 463.1699.

#### 3.1.2. General Procedure for the Synthesis of Target Compounds **3x**–**3aa**

Under a nitrogen atmosphere, R’-MgBr (1.0 M, 0.60 mL, 0.60 mmol, 3.0 equiv) was added dropwisely to a solution of **1a** or **1x** (0.20 mmol, 1.0 equiv) in anhydrous THF (3.0 mL) at 0 °C. After being stirred at 25 °C for 4 h, the reaction was quenched with water (15 mL) and extracted with EtOAc (10 mL × 3). The combined organic phase was dried over anhydrous Na_2_SO_4_, filtered, and concentrated to afford the crude diol, which was used in next step without further purification.

Under a nitrogen atmosphere, sodium hypochlorite pentahydrate (99.3 mg, 0.60 mmol, 3.0 equiv) was added to a solution of the above crude diol and tetra(*n*-butyl)ammonium hydrogen sulfate (13.6 mg, 0.04 mmol, 20 mol%) in DCM (2.0 mL) and water (0.5 mL) at rt. After stirring for 1 h, the mixture was quenched with water (10 mL) and extracted with CH_2_Cl_2_ (15 mL × 2). The combined organic layer was dried over anhydrous Na_2_SO_4_, filtered, and then concentrated in vacuo. The residue was purified by flash chromatography on silica gel (PE/EtOAc) to deliver the product **3x**–**3aa**.

The reaction of **1x** (69.2 mg, 0.20 mmol, 92% ee, 1.0 equiv) afforded product **3x** (69.3 mg, 82%, 89% ee, *S_T_* = 97%) (eluent for column chromatography on silica gel PE/EtOAc = 10:1) as a white solid.$\;{\left[\alpha \right]}_{D}^{20}$ + 88.3 (c 1.00, CH_2_Cl_2_). HPLC conditions: Chiralcel AD-H, isopropanol/hexane = 10:90, flow: 1.0 mL/min, λ = 210 nm, t_R_ = 16.4 min (minor), 30.8 min (major). **^1^H NMR** (500 MHz, CDCl_3_) δ 7.81–7.75 (m, 4H), 7.43–7.37 (m, 2H), 7.30–7.26 (m, 4H), 7.26–7.22 (m, 2H), 7.05 (dd, *J* = 8.0, 1.0 Hz, 2H), 6.88 (dd, *J* = 8.5, 1.0 Hz, 2H), 3.48 (s, 6H). ^**13**^**C NMR** (126 MHz, CDCl_3_) δ 196.4, 156.1, 139.7, 137.4, 132.2, 130.2, 128.0, 127.6, 124.3, 121.8, 112.8, 55.1. HRMS (ESI) calcd for C_28_H_22_O_4_Na [M+Na]^+^ 445.1410, found 445.1414.

The reaction of **1a** (62.8 mg, 0.20 mmol, 95% ee, 1.0 equiv) with EtMgBr (0.60 mmol, 3.0 equiv) afforded product **3y** (58.2 mg, 85%, 95% ee, *S_T_* = 100%) (eluent for column chromatography on silica gel PE/EtOAc = 10:1) as a white solid.$\;{\left[\alpha \right]}_{D}^{20}$ + 1.52 (c 1.00, CH_2_Cl_2_). HPLC conditions: Chiralcel OD-3, isopropanol/hexane = 2:98, flow: 0.8 mL/min, λ = 254 nm, t_R_ = 8.6 min (minor), 9.4 min (major). ^**1**^**H NMR** (500 MHz, CDCl_3_) δ 7.68–7.63 (m, 2H), 7.51–7.47 (m, 2H), 7.44 (d, *J* = 7.5 Hz, 1H), 7.39–7.34 (m, 2H), 7.34–7.29 (m, 2H), 7.27–7.25 (m, 1H), 7.25–7.22 (m, 1H), 2.80 (dq, *J* = 18.0, 7.0 Hz, 1H), 2.38 (dq, *J* = 18.0, 7.0 Hz, 1H), 2.03 (s, 3H), 2.01 (s, 3H), 0.91 (t, *J* = 7.0 Hz, 3H). **^13^C NMR** (126 MHz, CDCl_3_) δ 203.9, 197.2, 139.5, 138.1, 138.01, 137.98, 137.6, 137.4, 136.8, 132.7, 132.6, 131.8, 130.2, 128.0, 127.2, 126.6, 126.1, 125.7, 33.9, 20.2, 20.0, 8.0. HRMS (ESI) calcd for C_24_H_22_O_2_Na [M+Na]^+^ 365.1512, found 365.1519.

The reaction of **1a** (62.8 mg, 0.20 mmol, 95% ee, 1.0 equiv) with 1-methylvinylmagnesium bromide (0.60 mmol, 3.0 equiv) afforded product **3z** (53.9 mg, 76%, 96% ee, *S_T_* = 100%) (eluent for column chromatography on silica gel PE/EtOAc = 10:1) as a white solid.$\;{\left[\alpha \right]}_{D}^{20}$ + 7.06 (c 1.00, CH_2_Cl_2_). HPLC conditions: Chiralcel OD-3, isopropanol/hexane = 2:98, flow: 0.8 mL/min, λ = 254 nm, t_R_ = 7.3 min (minor), 8.2 min (major). ^**1**^**H NMR** (500 MHz, CDCl_3_) δ 7.65–7.61 (m, 2H), 7.51–7.45 (m, 1H), 7.42 (dd, *J* = 7.5, 1.5 Hz, 1H), 7.38–7.34 (m, 2H), 7.34–7.32 (m, 1H), 7.30–7.26 (m 1H), 7.24–7.19 (m, 2H), 7.14 (dd, *J* = 7.5, 1.5 Hz, 1H), 5.38–5.34 (m, 1H), 5.25–5.21 (m, 1H), 2.11 (s, 3H), 2.09 (s, 3H), 1.69 (s, 3H). ^**13**^**C NMR** (126 MHz, CDCl_3_) δ 199.1, 196.8, 143.9, 139.0, 138.8, 138.7, 137.9, 137.6, 137.1, 136.7, 132.5, 132.1, 131.4, 130.5, 130.1, 127.8, 127.1, 126.3, 126.13, 126.07, 20.12, 20.07, 17.3. HRMS (ESI) calcd for C_25_H_22_O_2_Na [M+Na]^+^ 377.1512, found 377.1524.

The reaction of **1a** (62.8 mg, 0.20 mmol, 95% ee, 1.0 equiv) with cyclopropylmagnesium bromide (0.60 mmol, 3.0 equiv) afforded product **3aa** (53.2 mg, 75%, 94% ee, *S_T_* = 99%) (eluent for column chromatography on silica gel PE/EtOAc = 10:1) as a white solid.$\;{\left[\alpha \right]}_{D}^{20}$ − 11.7 (c 1.00, CH_2_Cl_2_). HPLC conditions: Chiralcel IC-3, isopropanol/hexane = 2:98, flow: 0.8 mL/min, λ = 254 nm, t_R_ = 30.1 min (minor), 35.8 min (major). ^**1**^**H NMR** (500 MHz, CDCl_3_) δ 7.71–7.66 (m, 2H), 7.59 (dd, *J* = 7.5, 1.5 Hz, 1H), 7.52–7.47 (m, 1H), 7.43 (dd, *J* = 7.5, 1.5 Hz, 1H), 7.39–7.34 (m, 2H), 7.34–7.29 (m, 2H), 7.29–7.24 (m, 2H), 2.22–2.16 (m, 1H), 2.03 (s, 3H), 2.02 (s, 3H), 0.97–0.91 (m, 1H), 0.91–0.85 (m, 1H), 0.84–0.78 (m, 1H), 0.56–0.47 (m, 1H). ^**13**^**C NMR** (126 MHz, CDCl_3_) δ 204.2, 197.1, 139.7, 139.3, 138.2, 137.6, 137.4, 137.1, 137.0, 132.6, 132.5, 132.0, 130.4, 128.0, 127.2, 126.9, 126.2, 125.8, 20.2, 20.1, 20.0, 12.4, 11.0. HRMS (ESI) calcd for C_25_H_23_O_2_ [M + H]^+^, 355.1693, found 355.1687.

#### 3.1.3. Preparation of 2,2′-dimethyl-6,6′-bis(1-phenylvinyl)-1,1′-biphenyl **4**

Under a nitrogen atmosphere, a suspension of Ph_3_PMeBr (535.8 mg, 1.5 mmol, 7.5 equiv) in THF (5.0 mL) was added to *n-*BuLi (2.4 M in hexane, 0.62 mL, 1.5 mmol, 7.5 equiv) at 0 °C (ice bath), and the mixture was stirred at 0 °C for 30 min to yield a yellow mixture. A solution of ketone **3a** (78.0 mg, 0.20 mmol, 1.0 equiv) in THF (2.0 mL) was added at 0 °C. The reaction mixture was stirred at r.t. overnight. The resulting solution was quenched with aq. NH_4_Cl and extracted with ethyl acetate (3 × 10 mL). The combined organic phase was dried over anhydrous Na_2_SO_4_, filtrated, and concentrated in vacuo, and the residue was purified by column chromatography on silica gel (PE/EtOAc = 20:1) to give **4** (66.4 mg, 86%, 93% ee).$\;{\left[\alpha \right]}_{D}^{20}$ + 321 (c 1.00, CH_2_Cl_2_). HPLC conditions: Chiralcel IC, isopropanol/hexane = 0.1:99.9, flow: 0.8 mL/min, λ = 254 nm, t_R_ = 5.5 min (major), 8.1 min (minor). ^**1**^**H NMR** (500 MHz, CDCl_3_) δ 7.14 (dd, *J* = 7.5, 1.5 Hz, 2H), 7.12–7.05 (m, 4H), 7.05–6.99 (m, 4H), 6.83–6.79 (m, 4H), 6.79–6.76 (m, 2H), 5.16 (d, *J* = 1.5 Hz, 2H), 5.08 (d, *J* = 1.5 Hz, 2H), 1.55 (s, 6H). ^**13**^**C NMR** (126 MHz, CDCl_3_) δ 150.4, 142.5, 141.6, 137.9, 137.2, 128.6, 128.0, 127.2, 126.8, 126.8, 126.5, 117.0, 19.7. HRMS (ESI) calcd for C_30_H_27_ [M + H]^+^ 387.2107, found 387.2125.

#### 3.1.4. Preparation of 6,6′-dibenzoyl-[1,1′-biphenyl]-2,2′-dicarbaldehyde **5**

Under a nitrogen atmosphere, a mixture of **3a** (78.0 mg, 0.20 mmol, 1.0 equiv), benzoyl peroxide (24.2 mg, 0.10 mmol, 0.50 equiv) and NBS (356 mg, 2.0 mmol, 10 equiv) in CCl_4_ (10 mL) was stirred for reflux for 8 h. After being cooled to r.t., the resulting solution was quenched with water, and the mixture was extracted with CH_2_Cl_2_ (3 × 5.0 mL). The combined organic phase was dried over anhydrous Na_2_SO_4_, filtrated, and concentrated in vacuo to give a crude product, which was used without further purification.

The mixture of the above benzyl bromide and AgNO_3_ (552 mg, 2.0 mmol, 10 equiv) in CH_3_CN (2.0 mL) and H_2_O (1.0 mL) was stirred at 100 °C (oil bath) for 8 h. The resulting solution was cooled down and concentrated in vacuo. The residue was purified by column chromatography on silica gel to give **5** (51.1 mg, 61% for 2 steps, 93% ee).$\;{\left[\alpha \right]}_{D}^{20}$ + 7.02 (c 1.00, CH_2_Cl_2_). HPLC conditions: Chiralcel AD-H, isopropanol/hexane = 20:80, flow: 1.0 mL/min, λ = 254 nm, t_R_ = 12.2 min (minor), 16.1 min (major). ^**1**^**H NMR** (400 MHz, CDCl_3_) δ 9.81 (s, 2H), 8.22–8.09 (m, 2H), 7.66–7.57 (m, 4H), 7.54–7.46 (m, 4H), 7.45–7.36 (m, 2H), 7.23–7.16 (m, 4H). ^**13**^**C NMR** (126 MHz, CH_2_Cl_2_) δ 195.6, 190.4, 138.8, 137.9, 136.8, 136.4, 133.6, 133.3, 130.3, 130.2, 128.2, 128.0. HRMS (ESI) calcd for C_28_H_19_O_4_ [M + H]^+^ 419.1278, found 419.1281.

#### 3.1.5. Preparation of (6,6′-dimethyl-[1,1′-biphenyl]-2,2′-diyl)bis(diphenylmethanol) **6**

Under a nitrogen atmosphere, a mixture of **3a** (78.0 mg, 0.20 mmol, 1.0 equiv) in THF (5.0 mL) was added to PhLi (1.0 M in ether, 0.80 mL, 0.80 mmol, 4.0 equiv) at 0 °C (ice bath), and the mixture was stirred at 0 °C for 30 min and rt for 3 h. The resulting solution was quenched with aq. NH_4_Cl was extracted with ethyl acetate (3 × 10 mL). The combined organic phase was dried over anhydrous Na_2_SO_4_, filtrated, and concentrated in vacuo, and the residue was purified by column chromatography on silica gel (PE/EtOAc = 20:1) to give 6 (66.4 mg, 86%, 93% ee).$\;{\left[\alpha \right]}_{D}^{20}$ + 116 (c 1.00, CH_2_Cl_2_). HPLC conditions: Chiralcel OD-3, isopropanol/hexane = 1:99, flow: 0.7 mL/min, λ = 230 nm, t_R_ = 6.9 min (minor), 9.3 min (major). ^**1**^**H NMR** (500 MHz, CDCl_3_) δ 7.30–7.18 (m, 20H), 7.13–7.08 (m, 2H), 6.91–6.86 (m, 4H), 4.72 (s, 2H), 0.74 (s, 6H). ^**13**^**C NMR** (126 MHz, CDCl_3_) δ 148.9, 143.7, 143.2, 138.3, 138.2, 129.4, 128.8, 128.7, 128.0, 127.6, 127.5, 127.3, 126.8, 126.2, 84.3, 18.4. HRMS (ESI) calcd for C_40_H_34_O_2_Na [M + Na]^+^ 569.2451, found 569.2460.

#### 3.1.6. Free Radical Capture Experiment of **2a** with TEMPO

Under an air atmosphere, a solution of **2a** (39.2 mg, 0.10 mmol, 1.0 equiv) in THF (2 mL) was added to *t*BuOK (33.6 mg, 0.30 mmol, 3.0 equiv) and TEMPO (31.3 mg, 0.20 mmol, 2.0 equiv) at room temperature; then, the mixture was stirred at the same temperature for 30 min. A total of 0.5 mL of the reaction mixture was taken out and passed through a short pad of silica gel. The filtrate was analyzed by high-resolution mass. HRMS (ESI) calcd for C_37_H_42_NO_3_ [M + H]^+^ 548.3159, found 548.3168; calcd for C_37_H_41_NO_3_Na [M + Na]^+^ 570.2979, found 570.2992.

## 4. Conclusions

In conclusion, we have developed two sets of conditions for realizing oxidative C-C cleavage of dihydrophenanthrene-9,10-diols in the synthesis of axially chiral biaryl diketones. The merit of these two protocols is that the carbon–carbon bond cleavage ring-opening occurs under mild, metal-free conditions and in a very short reaction time, featuring a highly efficient point-to-axial chirality transfer process. Furthermore, the optically active diketones have been demonstrated to transform into an array of axially chiral compounds.

## Data Availability

The data presented in this study are available in the Supporting Information.

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
