# Peer review of "A Chiral Relay Race: Stereoselective Synthesis of Axially Chiral Biaryl Diketones through Ring-Opening of Optical Dihydrophenan-threne-9,10-diols"

_molecules, 2023, doi:10.3390/molecules28165956_

Round 1

Reviewer 1 Report

In this manuscript, Gu and co-workers reported the stereoselective synthesis of axially chiral biaryl ketones from the optically active dihydrophenan-threne-9,10-diols using a central-axial-chirality conversion strategy. Two sets of conditions were developed for oxidative cleavage of the C-C bond, resulting in the formation of axially chiral biaryl ketones with high enantiopurity retention. Moreover, the preliminary synthetic applications of the obtained axially chiral biaryl diketones were demonstrated by the authors. The work is clearly designed, and the conclusions are consistent with the experimental data. Consequently, I support the acceptance of this manuscript after the following issues been addressed.  

1) In Scheme 1a, the term of “Kinetic Dynamic resolution” should be corrected to “Dynamic Kinetic Resolution”

2) In Scheme 3b, the authors proposed that the hydroxy group adjacent to the phenyl group will preferentially attack the electrophilic chlorine. Is there any evidence supporting this regioselectivity?

3) The authors proposed a radical type mechanism for the tBuOk and oxygen mediated C-C bond cleavage. Is there any experimental evidence supporting this proposed mechanism?

In page 5, line 106, “atmospher” should be corrected to “atmosphere”

Reviewer 2 Report

In this publication, Gu and co-workers reported a stereospecific ring opening of enantiomerically enriched dihydrophenanthrene-9,10-diols through oxidative cleavage of a C-C bond to afford an interesting family of axially chiral diketones. This work emerges from previous results developed from the same research group. I think overall the work is of good interest to the scientific community and I recommend accepting the manuscript for publication after addressing the following points:

- Does the reaction work out well when the starting materials present an alkyl or alkenyl moiety in alpha position to the ketone, instead of an aryl group?

- “In previous work, we developed an efficient asymmetric arylation reaction of phenanthrene-9,10-diones for the preparation of optically active α-hydroxyl phenanthrenones.” Please indicate the reference of this previous work.

- The authors claim that radical species are generated during the reaction. This has not been checked out by any means experimentally. Can any reaction intermediate be trapped with radical scavengers (e.g. TEMPO) or through radical clock probes? Otherwise, I think the mechanistic proposal lacks of enough evidence to be stated with scientific soundness. Alternatively, if there are other precedents in the literature of a similar radical pathway for this C-C bond cleavage, please indicate it in the discussion of the mechanism.

- Some chromatograms of the HPLC traces do not present acceptable purity to be included in the SI (e.g. top chromatogram in page S61, top chromatogram in page S76). Please provide a better version of these chromatograms in the SI if possible.

Please check in detail again the manuscript to make sure the spelling of some words in English is correct. I could notice several typos while I was reading the manuscript, like: “which has been wide applications” (line 25), “diketons” (line 84), “Witting” (line 127

Round 2

Reviewer 2 Report

The authors have addressed all of the questions and issues that I had requested. I recommend to accept the article in its current form. Congratulations to the authors for such a nice work.